# Targeting Peripheral N-Methyl-D-Aspartate Receptor (NMDAR): A Novel Strategy for the Treatment of Migraine

**DOI:** 10.3390/jcm12062156

**Published:** 2023-03-10

**Authors:** Veberka Kalatharan, Mohammad Al-Mahdi Al-Karagholi

**Affiliations:** Danish Headache Center, Department of Neurology, Rigshospitalet Glostrup, Faculty of Health and Medical Sciences, University of Copenhagen, Valdemar Hansens Vej 5, DK-2600 Glostrup, Denmark

**Keywords:** glutamate, headache, pain, aura, CSD

## Abstract

**Backgrounds**: Several acute and preventive medications were developed for the treatment of migraine. Yet, a significant proportion of patients reports an inadequate response and a lack of tolerability, emphasizing the need for new options. Glutamate is the most important excitatory neurotransmitter in the brain, and glutamate receptors including N-Methyl-D-Aspartate Receptor (NMDAR) are expressed at several levels of the trigeminovascular system, which is the anatomical and physiological substrate of migraine pain. **Objective:** To review preclinical and clinical studies investigating the role of the NMDAR in migraine pathophysiology. **Methods:** No protocol was registered for this study. References for the present review were identified from a narrative search of the PubMed database. Search terms such as glutamate, migraine, N-Methyl-D-Aspartate Receptor, and NMDAR were used. No restrictions were made in terms of the language and date of publication. **Results:** In animal models, administration of monosodium glutamate (MSG) activated and sensitized trigeminovascular neurons. In healthy human participants, consumption of MSG caused headaches, craniofacial sensitivity, and nausea. In in vivo models and through immunolabeling, NMDAR subunits NR1, NR2A, and NR2B were expressed in trigeminal ganglion neurons. In humans, NMDAR antagonists such as ketamine and memantine caused a significant reduction in pain intensity and monthly headache frequency. **Conclusions:** Accumulative evidence indicates that NMDAR is a promising new target for the treatment of migraine. Selective NMDAR antagonists without central effects are needed to investigate their therapeutic benefit in the treatment of migraine.

## 1. Introduction

Migraine is a primary headache disorder, affecting more than 15% of the global adult population in their most productive years of life with a health and economic burden of billions of dollars globally [1]. The clinical manifestation of migraine is recurrent attacks with a severe, and usually unilateral and throbbing headache, lasting 4–72 h and associated with nausea and/or light and sound sensitivity [2]. In one-third of individuals with migraine, the headache phase is preceded by transient focal neurological disturbances, the so-called migraine aura-phase [3], whose underlying mechanism is considered to be cortical spreading depression (CSD) [4]. CSD is a self-sustaining, slowly propagating wave of transient neuronal and glial depolarization [3] that silences brain and electrical activity for several minutes [5,6,7]. In the past three decades, advances made in migraine research led to the development of several acute and preventive medications. Yet, a significant proportion of patients reports an inadequate response and a lack of tolerability [8]. 

Glutamate is the principal excitatory neurotransmitter in the central nervous system [9], and its role in migraine pathophysiology has been under the spotlight for more than three decades [10,11,12,13]. Glutamate receptors are pharmacologically classified as ionotropic and metabotropic receptors. N-Methyl-D-Aspartate Receptor (NMDAR) is an ionotropic glutamate receptor [9] that modulates excitatory neurotransmission by conducting sodium (Na^+^) and calcium (Ca^2+^) ions into the cell and potassium (K^+^) ions outside the cell. Key structures related to migraine pain, including the trigeminal ganglion (TG), trigeminal nucleus caudalis (TNC), and thalamus contain high densities of NMDAR-positive neurons, pinpointing a potential connection between the NMDAR and migraine pathophysiology [9,12,14,15,16]. 

Here, we provide a concise overview of the molecular structure and physiological function of NMDAR, highlight recent mechanistic insights into the role of this receptor in migraine pathophysiology, and discuss the possible complementarity and interdependence of signaling pathways involved in migraine attack initiation.

## 2. Methods

No protocol was registered for this study. References for the present review were identified through a narrative search of the PubMed database regarding glutamate and migraine on 27th of November 2022. The search terms *“glutamate signaling AND migraine”, “migraine AND glutamate AND target”, “Migraine AND glutamate”*, *“Migraine AND N-Methyl-D-Aspartate Receptor AND glutamate”* and *“Migraine AND NMDAR*” were used. No restrictions were made in terms of the language and date of publication. Additionally, references from relevant articles were identified. The final reference list was generated based on relevance to the topic by reading the title and abstract. 

## 3. Ion Channels in Migraine Pathophysiology

The origin of migraine pain involves activation and sensitization of trigeminal afferents innervating meningeal and large pial arteries, denoted as the trigeminovascular system (TVS) [2]. First-order neurons in the TG project nociceptive signals to second-order central neurons in the TNC [17] and axons from their relay to multiple nuclei in the brainstem, hypothalamus, and thalamus [18]. Using neuroimaging in spontaneous human migraine attacks, the hypothalamus was activated during the prodromal symptoms (changes in attention and concentration, mood swings, tiredness, frequent yawning, craving certain foods, and multiple others) which occur hours to days before headache onset in up to 90% of individuals with migraine [19]. Whether migraine pain is of peripheral provenance or whether hypothalamus and thalamus network to integrate peripheral influences with central mechanisms, facilitating the precipitation of migraine pain is yet to be elucidated. 

The human genome encodes at least 400 ion channel family members, which represent the second largest class of membrane proteins for drug discovery after G-protein-coupled receptors (GPCRs) [20]. Yet, ion channels remain considerably under-exploited as therapeutic targets [21]. Ion channels are expressed in migraine-related structures including cranial arteries, where they principally regulate the vascular tone. Dysfunction or abnormal regulation of ion channels might result in disruption of the excitation–inhibition balance, neuronal excitability, and peripheral and central sensitization [20,22]. Balance control impairment in neuronal function causes a heterogeneous group of disorders called channelopathies and includes epilepsy, episodic ataxia, and familial hemiplegic migraine (FHM) [23]. These findings and the observation that migraine is more prevalent in individuals with channelopathies including epilepsy and episodic ataxia [24] indicate that migraine might be a channelopathy disease.

## 4. Glutamate and Migraine

The role of glutamate amino acid in migraine has been investigated in human models in the past three decades. Glutamate plays a role in neurogenic inflammation and excitation of the trigeminal pain pathway. All classes of glutamate receptors are located in superficial laminae I and II of the trigeminocervical complex (TCC) [12,18], which forms the base of the complex interaction and modulation of neuronal pain signals. Moreover, extracellular glutamate concentrations increase during CSD [25,26], and this increase further charges CSD and facilitates its propagation by activating NMDARs [27,28,29]. Ferrari and colleagues showed that individuals with migraine with aura had higher plasma glutamate levels compared to healthy participants, individuals with migraine without aura, and individuals with tension-type headache [10]. Plasma glutamate levels were further increased during attacks in individuals with migraine with and without aura [10]. D’Andrea and colleagues showed that, platelet glutamate levels (platelet dense granules carry a considerable amount of glutamate) were significantly higher in individuals with migraine with aura compared to healthy participants and other headache groups, such as individuals with migraine without aura, or with tension headaches and cluster headaches [30]. 

Additional studies supported that individuals with migraine with and without aura had higher plasma glutamate levels compared to healthy participants [31,32]. Interestingly, migraine preventive therapies were found to reduce plasma glutamate levels [32]. The involvement of glutamate signaling in headache and migraine was further reinforced by the finding that (1) the consumption of a single dose (150 mg/kg) of monosodium glutamate (MSG) caused headache, craniofacial sensitivity, and nausea in healthy participants [33], and (2) repeated MSG intake (150 mg/kg) for five daily sessions for one-week reduced pressure pain thresholds and caused headache in healthy participants compared to placebo [34]. 

## 5. N-Methyl-D-Aspartate Receptor (NMDAR)

Glutamate receptors are pharmacologically classified as ionotropic (ligand-gated ion channels) and metabotropic (GPCRs). NMDAR along with alpha-amino-3-hydroxy-5-methyl-4-isoxazole-propionic acid receptor (AMPAR) and Kainate receptor (KAR) are categorized as ionotropic glutamate receptors [9]. NMDAR modulates excitatory neurotransmission by conducting Na^+^ and Ca^2+^ ions into the cell and K^+^ ions outside the cell (Figure 1). 

The NMDAR is a heterotetrameric complex assembled from three different NMDAR subunits (NR1, NR2, and NR3) [35] (Figure 2). The NR1 subunit is essential for a functional NMDAR complex [35,36] and is represented by eight splice variants, which influence the channel properties. Four NR2 subunits (NR2A, NR2B, NR2C and NR2D) and two NR3 subunits (NR3A and NR3B) have been mapped. The NR1-NR2A heterodimer suggests a mechanism for ligand-induced ion channel opening and is the functional unit in tetrameric NMDARs [37]. Table 1 presents the predominant localization of NMDAR subunits. All NMDAR subunits share a common membrane structure characterized by a large N-terminus domain (NTD), an agonist/ligand binding domain (ABD/LBD), a transmembrane region (TMD), and a cytoplasmic region (CTD) (Figure 2) [35]. NMDAR activation requires the simultaneous binding of glycine and glutamate, together with the removal of the endogenous channel-pore blocker, magnesium ion (Mg^2+^), in a voltage-dependent manner [38,39]. The particular importance of the NMDAR is due to the high permeability to Ca^2+^ ions that gives NMDARs a significant role in both synaptic plasticity under physiological conditions and neuronal death under excitotoxic pathological conditions [35]. A typical NMDAR complex consists of two glycine-binding NR1 subunits and two glutamate-binding NR2 subunits [36]. Incorporating either NR3A or NR3B subunits showed decreased channel activity via reduced single-channel conductance, reduced Ca^2+^ permeability and increased Mg^2+^ blockade [35,40,41,42]. Notably, the NR3 subunit can bind to glycine rather than glutamate, which is similar to the NR1 subtype. Thus, a novel type of NMDAR complex composed of NR1 and NR3 subunits would only require glycine and not glutamate for activation [43]. The entry of Ca^2+^ into dendritic spines through NMDARs is essential for long-term potentiation (LTP) and long-term depression (LTD) [44]. LTP is induced by a high-frequency synaptic activity that causes postsynaptic membrane depolarization, a decrease in voltage-dependent Mg^2+^ blockage of the NMDAR pore, and a massive entry of Ca^2+^ ions into dendritic spines leading to calmodulin (CaM) and CaM-dependent kinase II activation (Figure 1).

**Figure 1 jcm-12-02156-f001:**
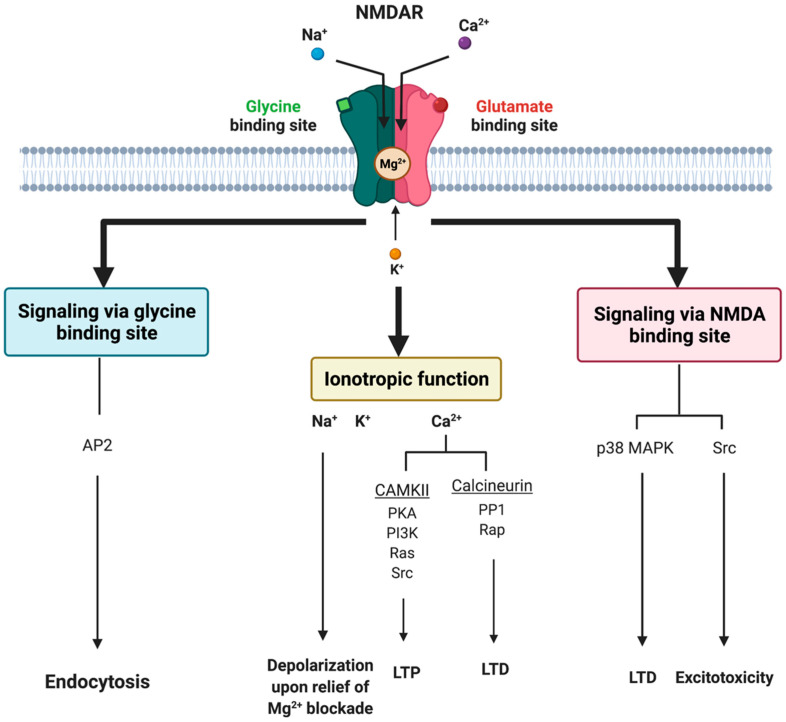
**Tripartite signaling of N-Methyl-D-Aspartate Receptor (NMDAR).** This model illustrates three parallel signaling pathways of NMDAR transduction. Binding of glutamate and glycine results in ionotropic function and membrane depolarization through efflux of K^+^ ions and influx of Na^+^ and Ca^2+^ ions mediating downstream Ca^2+^-dependent pathways. NMDAR signaling can also happen non-ionotropically [45] when glycine or glutamate bind independently and induce conformational changes and downstream protein interactions. Glycine stimulation enhances NMDAR association with the AP2 and subsequent activation of a downstream endocytic pathway. Lastly, glutamate binding can result in LTD through p38 MAPK and excitotoxicity though an activation of Src. AP2 = endocytic adaptor protein 2, CaMKII = calcium/calmodulin kinase II, LTD = long-term depression, LTP = long-term potentiation, p38 MAPK = p38 mitogen-activated protein kinase, PP1 = protein phosphatase 1, Rap = ras-related protein, Src = thyrosin kinase family.

### 5.1. Evidence Implicating NMDAR in Migraine 

NMDARs are expressed in trigeminovascular neurons (trigeminal neurons innervating dural blood vessels) [47]. Glutamate transporter (excitatory amino acid transporter-2 (*EAAT-2*)) was identified in dural blood vessels [48]. In rat models, the expression of NMDAR subunits NR1, NR2A, and NR2B have been demonstrated in TG [49,50,51,52]. Specifically, the NR2B subunit was expressed in nerve fibers that innervated dural blood vessels [53]. In a recent study, Guerrero-Toro and colleagues investigated the role of peripherally released glutamate in trigeminal nociception by using TG neurons from rodent models. The authors revealed (1) the presence of NR2A and NR2B subunits in TG neurons through immunolabeling, (2) the expression of other glutamate receptors in TG neurons, (3) the presence of calcitonin gene-related peptide (CGRP) enhanced the fraction of NMDAR-positive neurons, and (4) the removal of Mg^2+^ increased the nociceptive firing of trigeminal afferents by NMDAR in in vivo conditions [54]. Moreover, NMDAR activation mediated nociceptive transmission in the TNC [55], and the stimulation of dural structures caused a co-release of glutamate and CGRP in the TCC [56]. In rats, intravenous administration of 50 mg/kg MSG activated and sensitized the trigeminovascular neurons associated with dural vasodilation. Co-administration of a peripherally restricted NMDAR antagonist, (2R)-amino-5-phosphonovaleric (APV) attenuated the MSG-induced effects [53]. Glutamate is a negatively charged, polar amino acid. The central uptake of glutamate and other anionic excitatory amino acids from the peripheral circulation is limited by the blood–brain barrier (BBB). At physiologic plasma concentrations, glutamate flux from plasma into the brain is mediated by a high-affinity transport system at the BBB. Efflux from the brain back into plasma appears to be driven in large part by a Na^+^-dependent active transport system at the capillary abluminal membrane. The glutamate concentration in brain interstitial fluid is only a fraction of that of plasma and is maintained fairly independently of small fluctuations in plasma concentration [57]. Thus, glutamate-induced headache is likely due to peripheral activation of NMDAR, indicating that targeting peripheral NMDAR might be a possible strategy for the treatment of individuals with migraine to avoid side effects due to central action.

### 5.2. NMDAR Antagonists in Preclinical Studies

Based on the knowledge about NMDAR subunits and their distinctive domains, several potential therapeutic approaches such as allosteric modulators, and competitive and non-competitive antagonists have been proposed for anti-migraine therapy (Figure 2). The negative allosteric modulators of NMDAR include the endogenous zinc ion (Zn^2+^) and ifenprodil-like compounds [35,58], which both act on the NR2 NTD as non-competitive antagonists. Zn^2+^ has been shown to be a selective antagonist of the subunits NR2A and NR2B, whereas ifenprodil and its derivatives are selective antagonists of NR2B with more than a 100-fold higher affinity [59]. The pore domain of the NMDAR forms binding sites for non-competitive pore blockers such as the endogenous Mg^2+^ and exogenous compounds such as MK-801 (also known as dizocilpine), memantine, and ketamine. Memantine and Mg^2+^ are low-affinity NMDAR blockers with faster blocking and unblocking kinetics, which are linked to fewer and less severe side effects than substances with slower receptor kinetics such as ketamine and MK-801 [38]. The administration of dizocilpine maleate, an NDMAR antagonist, resulted in a substantial blockade of neuronal firing and the inhibition of trigeminovascular-evoked responses in the trigeminovascular complex in cats [47]. Apart from being involved in the trigeminal pain pathway, the NMDAR seems to be implicated in the initiation and propagation of CSD, which is believed to be the pathophysiological mechanism underlying migraine aura [5,55,60]. Mg^2+^ selectively suppressed glutamate-induced CSD [61], and MK-801 (non-specific NMDAR antagonist) and ifenprodil (NR2B selective antagonist) inhibited experimental-induced CSD [27,60,62,63] (Box 1). In 2019, Hoffmann and colleagues investigated the efficacy of Mg^2+^ and memantine in rat models by comparing microiontophoretic application and systemic administration to modulate trigeminovascular nociception. They observed that intravenous systemic administration of moderate concentrations of low-affinity NMDAR channel blockers such as Mg^2+^ and memantine does not inhibit stimulus-evoked firing significantly compared to microiontophoretic application [38]. 

Box 1Preclinical evidence.
Glutamate N-Methyl-D-Aspartate Receptor (NMDAR) is expressed in nerve fibers innervating dural blood vessels. Glutamate transporter (excitatory amino acid transporter-2 (*EAAT-2*)) is expressed in dural blood vessels [48].In rats, intravenous administration of 50 mg/kg monosodium glutamate (MSG) activated and sensitized trigeminovascular neurons associated with dural vasodilation. Co-administration of a peripherally restricted NMDAR antagonist (2R)-amino-5-phosphonovaleric (APV) attenuated MSG-induced effects [53].Intraperitoneal injections of 1000 mg/kg MSG caused headache-like and nausea symptoms in association with an increase in plasma calcitonin-gene related peptide (CGRP) concentration. MSG-induced headache-like symptoms were inhibited by APV and partly attenuated by olcegepant, a selective CGRP receptor antagonist [64].Administration of dizocilpine maleate, a NDMAR antagonist, and GYKI 52466, a non-NMDAR antagonist, resulted in a substantial showed blockade of neuronal firing and inhibition of trigeminovascular-evoked responses in the trigeminovascular complex in cats [47].


### 5.3. NMDAR Antagonists in Clinical Studies

NMDAR antagonists such as Mg^2+^, ketamine, and memantine have been investigated for migraine treatment and prophylaxis, respectively [65]. Administration of Mg^2+^ has been examined in numerous clinical trials either perorally or intravenously [66]. In a prospective quasi-experimental study, Baratloo and colleagues found that intravenous magnesium sulfate might be superior to intravenous caffeine for the acute management of migraine headaches [67]. A double-blinded, cross-over multicenter pilot study reported a significant reduction in the incidence of migraine attacks when administering peroral triagnesium dietrate compared to placebo [68] (Table 2). Additionally, the administration of transdermal MgCl_2_ solution is described as an ideal approach in acute migraine treatment due to the immediate absorption through the skin [66,69]. Collectively, Mg^2+^ seems to have beneficial effect for the acute and preventive treatment of migraine. 

Several studies have investigated the use of ketamine or AMPA (LY293558; BGG492) antagonists as abortive therapies in migraine with aura and familial hemiplegic migraine [71,73,77]. Although most of these studies concluded that these drugs possessed limited or at best moderate efficacy in the reduction in severity of headaches and analgesic consumption, the presence of side effects owing to their effects on central glutamate receptors limited future exploration of their use in migraine. Regarding migraine prophylaxis, the efficacy of memantine has been investigated in retrospective (in 2007, [65]), open-label (in 2008, [75]), and randomized double-blinded placebo-controlled studies (in 2016, [76]). The participants in the three mentioned studies were individuals with migraine with and without aura as well as individuals with refractory migraine who received peroral doses of memantine 5 to 20 mg/day (Table 2). Together, these studies reported a memantine-induced significant reduction in (1) the monthly headache frequency, (2) the mean number of days with severe pain, and (3) the mean disability score. 

For the treatment of migraine with severe and prolonged aura, ketamine was studied in a small open-label study (*n* = 11) (in 2000, [71]) and a randomized double-blinded placebo-controlled trial (in 2013 [72] and 2018 [74]). The intranasal administration of ketamine (25 mg) reduced the severity, and to a lesser extent, the duration of aura [71,72]. However, intravenous administration of a low-dose (0.2 mg/kg) of ketamine for the treatment of acute migraine did not produce a significant reduction in pain or disability-related scores [74]. In 1995, Nicolodi and colleagues investigated the administration of subcutaneous ketamine hydrochloride in individuals with migraine and stated that there was noticeable pain relief in the ketamine group compared to the placebo group [70]. In 2016, Lauritsen and colleagues reported in a retrospective study that six out of six individuals with refractory migraine achieved the target pain relief endpoint with intravenous ketamine infusion [73] (Table 2 and Box 2). Collectively, it can be assumed that migraine with aura involves glutamate release and NMDAR activation, which could initiate migraine attacks [16].

The most common side effects reported from the above-mentioned trials with ketamine and memantine are asthenia, dizziness, nausea, psychiatric symptoms such as hallucinations, rashes, cognitive dysfunctions, somnolence, and changes in cardiovascular and respiratory stability [65,70,73,75]. So far, the American Headache Society [78,79] and European Headache Federation [80] have not recommended ketamine or memantine for the treatment of migraine. Mg^2+^, on the contrary, takes place as a nutraceutical in the migraine prevention treatment [78,80,81].

Box 2Clinical evidence.
Genome-wide association studies have shown significant polymorphisms in glutamate receptor and transporter genes that are associated with migraine [82].Consumption of a single dose (150 mg/kg) of monosodium glutamate (MSG) caused headache, craniofacial sensitivity, and nausea in healthy participants [33].Repeated MSG intake (150 mg/kg) in five daily sessions for one week reduced pressure pain thresholds and caused headaches in healthy participants compared to placebo [34].Glutamate levels measured in plasma and cerebrospinal fluid were elevated both ictally and interictally in individuals with migraine compared to healthy controls [10], and migraine preventive therapies reduced plasma glutamate levels [32].N-Methyl-D-Aspartate Receptor (NMDAR) antagonist drugs such as ketamine and memantine showed some efficacy in acute and prophylaxis migraine treatment [65,71,73,77].


## 6. Perspective and Conclusions

Glutamate and NMDAR are involved in the initiation of migraine aura and migraine pain. NMDAR subunits expressed in the trigeminal afferents are involved in trigeminal nociception. NMDAR antagonists such as memantine and ketamine might be applicable for individuals with migraine with aura [62] despite some studies showing low efficacy, which can be explained by their central mechanism of action. As migraine is increasingly being recognized as a heterogeneous disorder, available NMDAR antagonists might have beneficial effects in a subgroup of individuals with migraine, including in individuals with difficult-to-treat migraine. So far, ketamine or memantine have not been recommended in the treatment of migraine [78,79,80]. Mg^2+^ (NMDAR blocker), on the contrary, takes place as a nutraceutical in migraine prevention treatment [78,80,81]. More selective antagonists targeting peripheral NMDAR are needed to explore their beneficial effects in the treatment of migraine.

## Figures and Tables

**Figure 2 jcm-12-02156-f002:**
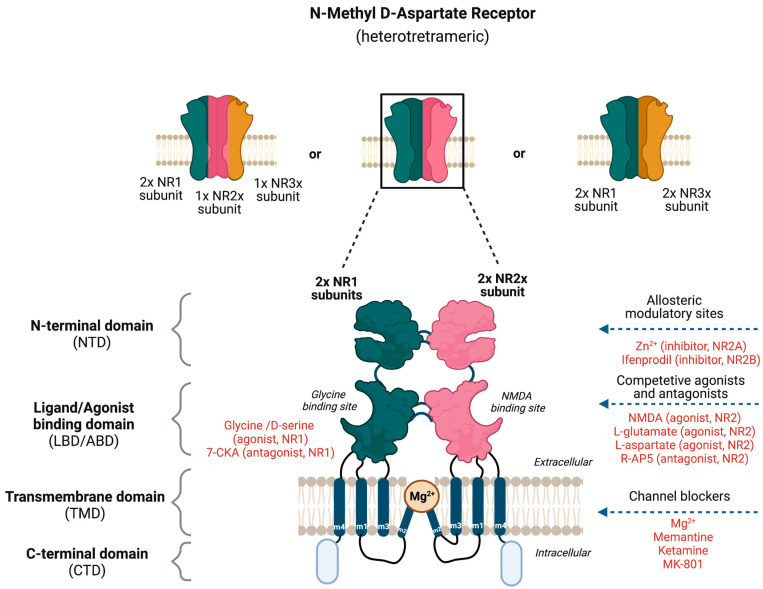
**Potential binding sites for ligands that modulate the activity of N-Methyl-D-Aspartate Receptor (NMDAR).** The NMDAR is assembled as tetramers; two NR1 and two NR2 subunits forming a dimer–dimer structure or two NR1 subunits associating with one NR2 and NR3 subunit. The channel construction with different domains gives rise to ligand binding or allosteric modulatory sites that either work as an NMDAR agonist or antagonist.

**Table 1 jcm-12-02156-t001:** Predominant localization of the NMDAR subunits.

NMDAR Subtype	Predominant Localization	References
NR1	Present throughout CNS.	[9]
NR2A	Thalamus; prominently in the lateral thalamic nuclei.	[9]
NR2B	Thalamus.	[9]
NR2C	High expression in cerebellum, low elsewhere.	[9]
NR2D	High expression early during development, low expression in adults.	[9]
NR3A	Spinal cord, thalamus, hypothalamus, brainstem, hippocampus (CA1), amygdala, and certain parts of cortex cerebri.	[40,46]
NR3B	In adults: expressed in somatic motor neurons in the brainstem and spinal cord, cerebellum, and hippocampus. The distribution of NR3B seems to be as ubiquitous as NR1.	[40,41,42,47]

**Table 2 jcm-12-02156-t002:** Clinical studies investigating NMDAR antagonists.

NMDAR Antagonist	Author and Year	Study Design	Number of Participants	Dose (Administration Form)	Type of migraine (With or Without Aura)	Findings
**Magnesium**	Baratloo A, et. al. (*2017*) [67]	Prospective quasi-experimental study	Total *n* = 70 Completed study *n* = 70	2 g Magnesium Sulfate (*intravenous*)	Individuals with migraine (Aura not reported)	Intravenous magnesium sulfate (2g) might be superior to intravenous caffeine (60 mg) for the short-term management of migraine headaches.
**Magnesium**	K. Taubert, et. al. (*1994*) [68]	Double-blinded, cross-over multicenter pilot study	Total *n* = 43 Completed study *n* = 43	600 mg/day Trimagnesium Dieitrate (*oral*)	Individuals with migraine (Aura not reported)	Significant reduction in the incidence of migraine attacks was observed.
**Ketamine**	Nicolodi et. al. (*1995*) [70]	Randomized double- blinded, cross-over study	Total *n* = 34 Completed study *n* = 34	80 µg/kg Ketamine hydrochloride (*subcutaneous*)	Individuals with migraine (Aura not reported)	Compared to the placebo group, Individuals in the ketamine group experienced a marked relief of pain both as an acute and prophylactic treatment.
**Ketamine**	Kaube et. al. (*2000*) [71]	Open label study	Total *n* = 11 Completed study *n* = 11	25 mg (*Intranasal*)	Familial hemiplegic migraine	5 out of 11 participants experienced a reduction in severity and duration of migraine attacks after ketamine administration, whereas 6 out of 11 participants had no beneficial effect.
**Ketamine**	Afridi et. al. (*2013*) [72]	Randomized double-blinded placebo-controlled trial	Total *n* = 30 Completed study *n* = 18	25 mg (*Intranasal*)	Individuals with migraine (With aura)	Intranasal ketamine is effective at reducing aura severity in individuals with prolonged aura.
**Ketamine**	Lauritsen et. al. (*2016*) [73]	Retrospective study	Total *n* = 6 Completed study *n* = 6	0.1 mg/kg (*Intravenous*)	Refractory migraine	6 out of 6 participants obtained the target pain relief endpoint after infusion of ketamine. The mean ketamine infusion rate at the time of pain relief was 0,34 mg/kg/hour.
**Ketamine**	Etchinson et. al. (*2018*) [74]	Randomized double-blinded placebo-controlled trial	Total *n* = 34 Completed study *n* = 34	0.2 mg/kg (*Intravenous*)	Individuals with migraine (With and without aura)	The difference in NRS pain scores was neither statistically nor clinically significant between ketamine group (*n* = 16) and placebo group (*n* = 18).
**Memantine**	Charles et. al. (*2007*) [65]	Retrospective study	Total *n* = 60 Completed study *n* = 54	5 mg to 20 mg/day (*Peroral*)	Individuals with migraine (With and without aura)	67% reported a greater than 50% reduction in estimated monthly headache frequency.
**Memantine**	Bigal et. al. (*2008*) [75]	Prospective, open label study	Total *n* = 38 Completed study *n* = 23	10 mg to 20 mg/day (*Peroral*)	Refractory migraine	A significant reduction in headache frequency, severity, and MIDAS score was observed after 3 months of treatment.
**Memantine**	Noruzzadeh et. al. (*2016*) [76]	Randomized double-blinded placebo-controlled trial	Total *n* = 52 Completed study *n* = 52	10 mg/day (*Peroral*)	Individuals with migraine (Without aura)	Individuals in the memantine group (*n* = 25) showed significantly greater reduction in monthly headache attack frequency and headache severity than the placebo group (*n* = 27).

## Data Availability

The study did not report any data.

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
