# Peer review of "Targeting Peripheral N-Methyl-D-Aspartate Receptor (NMDAR): A Novel Strategy for the Treatment of Migraine"

_jcm, 2023, doi:10.3390/jcm12062156_

Round 1

Reviewer 1 Report

Congratulation to the authors of the review entitled "Targeting Peripheral N-Methyl-D-Aspartate Receptor (NMDAR): A novel strategy for the treatment of migraine" submitted for possible publication in JCM. The paper is clear, well written and well balanced between fondamental evidence and clinical perspectives on the topic. I have few comments to address to the authors :

1) Although the authors clearly explained that activation of NMDAR ionotropic pathway requires the removal of Magnesium ion, they did not review the evidence of Magnesium in prevention of migraine, which has been studied in several studies. Why? Magnesium deserves to be included in the review. If the authors think that the mechanism of Magnesium in prevention on migraine is unclear and multifactorial, they should state it clearly and write a couple of sentences to explain.

2) Abstract, line 24 "NMDA antagonists..." : please precise whether it is in humans with migraine.

3) Introduction, lines 53-54 "Yet, a significant proportion..." : please add a reference.

4) Section 5.3. line 228 "peroral doses of ketamine 5 to 20 mg/day" : Did they authors mean memantine instead of ketamine?

5) Section 5.3. second paragraph : could the authors report the side effects of memantine and ketamine and whether guidelines have recommended these treatments in migraine ?

6) Table 2 would benefit from another column indicating the number of participants included in the studies. It will show how the studies did not allow for a definitive conclusion.

Author Response

Reviewer #1

Comments for the Author

Congratulation to the authors of the review entitled "Targeting Peripheral N-Methyl-D-Aspartate Receptor (NMDAR): A novel strategy for the treatment of migraine" submitted for possible publication in JCM. The paper is clear, well-written, and well-balanced between fundamental evidence and clinical perspectives on the topic. I have a few comments to address to the authors:

  1. Although the authors clearly explained that activation of NMDAR ionotropic pathway requires the removal of Magnesium ion, they did not review the evidence of Magnesium in prevention of migraine, which has been studied in several studies. Why? Magnesium deserves to be included in the review. If the authors think that the mechanism of Magnesium in prevention on migraine is unclear and multifactorial, they should state it clearly and write a couple of sentences to explain.

Response: Thank you for your comment. This is added in sections 5.2 and 5.3.

  1. Abstract, line 24 "NMDA antagonists...": please precise whether it is in humans with migraine.

Response: The precision is added in the abstract.

  1. Introduction, lines 53-54 "Yet, a significant proportion...": please add a reference.

Response: Reference is added.

  1. Section 5.3. line 228 "peroral doses of ketamine 5 to 20 mg/day" : Did they authors mean memantine instead of ketamine?

Response: This is corrected.

  1. Section 5.3. second paragraph: could the authors report the side effects of memantine and ketamine and whether guidelines have recommended these treatments in migraine?

Response: Both suggestions are included.

  1. Table 2 would benefit from another column indicating the number of participants included in the studies. It will show how the studies did not allow for a definitive conclusion.

Response: Thank you for your comment. We agree, and this is included.

Reviewer 2 Report

In this review article, the authors describe the implications of glutamate receptors, especially the NMDA receptor, in migraine. The contents of the article are concise and easy to understand. This is an excellent review article providing the overview of the role of glutamate receptors in migraine. 

Author Response

Reviewer #2

Comments for the Author

In this review article, the authors describe the implications of glutamate receptors, especially the NMDA receptor, in migraine. The contents of the article are concise and easy to understand. This is an excellent review article providing the overview of the role of glutamate receptors in migraine. 

Response: Thank you for your comment. We appreciate your time and effort.

Reviewer 3 Report

The role of glutamate and its receptors in migraine pathophysiology and as a site of a potential pharmacological intervention has outmost importance, thus the topic is interesting.

Unfortunately there is a number of reviews already published on this subject (glutamate + migraine), thus it is hard to find novelties and the content of the manuscript should be vastly improved, mainly with restructuration and adding further experimental data.

Remarks:

Title:

It speaks about peripheral NMDA receptors, in the paper there is no/little description of the role NMDA receptors outside the CNS and their possible modulation.

Introduction:

The key aspects of migraine pathophysiology should be added to the first part of the introduction: hypothalamic activation possibly related to the prodrome, CSD and activity of the migraine generators, activation and sensitization of the trigeminal system as the culprit of the attack. Later the effects of glutamate/its receptors should be discussed in light of these migraine phases.

Ion channels and migraine pathophysiology:

This part of the paper is too general does not contain enough plain scientific facts about the role of ion channels in migraine pathophysiology, especially in the context of the topic (eg. CSD generation), nor in the field other important structures (TRPV1, K channels). But frankly I don’t believe that this section is necessary at all.

Glutamate in migraine:

Besides discussing glutamate levels in migraine patients and the effect of supplementation the authors should briefly describe the role of glutamate in trigeminal nociception and sensitization as well. Moreover there is possible evidence that genetic alteration of glutamate neurotransmission is also linked with migraine. Last but not least glutamate plays an important role in the generation of CSD, which should be also described.

NMDAR

This section is vastly overpresented, although it does not contain novelties or facts directly linked to the topic (migraine pathogenesis). For the sake of the structure of the manuscript it cannot be skipped, but it should be shortened.

NMDAR antagonists in preclinical trials:

The effects of kynurenic acid and its derivatives in various headache models should be included. It might be also of relevance since KYNA penetrates poorly the BBB and probably acts on the peripheral arm of the glutamatergic system. Also the results with MK801 and memantine in animal studies should be presented in a more detailed way. The part describing the mechanism of action of particular antagonists, especially those not included in experimental or clinical studies (eg Zn) should be put in in the NMDAR section (5.)

NMDAR antagonists in clinical trials :

The last paragraph should rather go to the conclusions.

Conclusions:

To brief and to general.

Author Response

Reviewer #3

Comments for the Author

The role of glutamate and its receptors in migraine pathophysiology and as a site of a potential pharmacological intervention has outmost importance, thus the topic is interesting.

Unfortunately, there is a number of reviews already published on this subject (glutamate + migraine), thus it is hard to find novelties and the content of the manuscript should be vastly improved, mainly with restructuration and adding further experimental data.

Remarks:

  1. Title: It speaks about peripheral NMDA receptors, in the paper there is no/little description of the role NMDA receptors outside the CNS and their possible modulation.

Response: Thank you for your comment. We have on several occasions described the location and modulation of NMDA receptors outside the CNS. The following points from the present review stating the role of peripheral NMDA receptors in peripheral nerve fibers, trigeminal ganglion, trigeminal nucleus caudatis, etc. (from sections 5.1 and 5.2):

  • “Specifically, the NR2B subunit was expressed in nerve fibers that innervated dural blood vessels 1 and TG.”
  • Co-localization of CGRP and NMDAR: “2) the presence of calcitonin gene-related peptide (CGRP) enhanced the fraction of NMDA positive neurons.”
  • “Moreover, NMDAR activation mediated nociceptive transmission in the TNC 2
  • “Stimulation of dural structures caused co-release of glutamate and CGRP in the trigeminocervical complex (TCC).”
  • “In rats, intravenous administration of 50 mg/kg monosodium glutamate (MSG) activated and sensitized trigeminovascular neurons associated with dural vasodilation.”
  • “Co-administration of a peripherally restricted NMDA receptor antagonist, (2R)-amino-5-phosphonovaleric (APV) attenuated MSG-induced effects 1.”
  • “consumption of a single dose (150 mg/kg) of monosodium glutamate (MSG) caused headache, craniofacial sensitivity, and nausea in healthy volunteers 3
  • “Repeated MSG intake (150 mg/kg) in 5 daily sessions for one-week reduced pressure pain thresholds and caused headaches in healthy volunteers compared to placebo 4.”

We agree that there is a number of already published reviews implicating glutamate in migraine pathophysiology and it is important to highlight that this is not the theme of the present review. Here, we tried to capture the role of peripheral NMDA receptor in migraine pathophysiology.

  1. Introduction: The key aspects of migraine pathophysiology should be added to the first part of the introduction: hypothalamic activation possibly related to the prodrome, CSD and activity of the migraine generators, activation and sensitization of the trigeminal system as the culprit of the attack. Later the effects of glutamate/its receptors should be discussed in light of these migraine phases.

Response: These are added in section 3.  

  1. Ion channels and migraine pathophysiology: This part of the paper is too general does not contain enough plain scientific facts about the role of ion channels in migraine pathophysiology, especially in the context of the topic (eg. CSD generation), nor in the field other important structures (TRPV1, K channels). But frankly I don’t believe that this section is necessary at all.

Response: Thank you for your comment. We modified this section.

  1. Glutamate in migraine: Besides discussing glutamate levels in migraine patients and the effect of supplementation the authors should briefly describe the role of glutamate in trigeminal nociception and sensitization as well. Moreover, there is possible evidence that genetic alteration of glutamate neurotransmission is also linked with migraine. Last but not least glutamate plays an important role in the generation of CSD, which should be also described.

Response: The suggestions are added.

  1. NMDAR: This section is vastly overpresented, although it does not contain novelties or facts directly linked to the topic (migraine pathogenesis). For the sake of the structure of the manuscript it cannot be skipped, but it should be shortened.

Response: Thank you for your comment. Since NMDAR is the main focus of the present review, we in this section highlighted facts that have been used throughout the manuscript to give the reader a propriate understanding of the subject. 

  1. NMDAR antagonists in preclinical trials: The effects of kynurenic acid and its derivatives in various headache models should be included. It might be also of relevance since KYNA penetrates poorly the BBB and probably acts on the peripheral arm of the glutamatergic system. Also, the results with MK801 and memantine in animal studies should be presented in a more detailed way. The part describing the mechanism of action of particular antagonists, especially those not included in experimental or clinical studies (eg Zn) should be put in in the NMDAR section (5.)

Response: Thank you for your comment. We conducted a phase 1 ascending dose study, where we for the first time in human history administrated intravenous l-kynurenine to healthy participants. However, we observed no headache or other vascular effects which were previously reported in preclinical studies 5.

  1. NMDAR antagonists in clinical trials: The last paragraph should rather go to the conclusions.

Response: Thank you for your comment. We modified this section and the conclusion.

  1. Conclusions: To brief and to general.

Response: Conclusion is revised.

References

  1. O'Brien M and Cairns BE. Monosodium glutamate alters the response properties of rat trigeminovascular neurons through activation of peripheral NMDA receptors. Neuroscience 2016; 334: 236-244. 20160811. DOI: 10.1016/j.neuroscience.2016.08.007.
  2. Wang XM and Mokha SS. Opioids modulate N-methyl-D-aspartic acid (NMDA)-evoked responses of trigeminothalamic neurons. J Neurophysiol 1996; 76: 2093-2096. 1996/09/01. DOI: 10.1152/jn.1996.76.3.2093.
  3. Baad-Hansen L, Cairns B, Ernberg M, et al. Effect of systemic monosodium glutamate (MSG) on headache and pericranial muscle sensitivity. Cephalalgia 2010; 30: 68-76. DOI: 10.1111/j.1468-2982.2009.01881.x.
  4. Shimada A, Cairns BE, Vad N, et al. Headache and mechanical sensitization of human pericranial muscles after repeated intake of monosodium glutamate (MSG). J Headache Pain 2013; 14: 2. 20130124. DOI: 10.1186/1129-2377-14-2.
  5. Al-Karagholi MA, Hansen JM, Abou-Kassem D, et al. Phase 1 study to access safety, tolerability, pharmacokinetics, and pharmacodynamics of kynurenine in healthy volunteers. Pharmacol Res Perspect 2021; 9: e00741. DOI: 10.1002/prp2.741.
  6. Hautakangas H, Winsvold BS, Ruotsalainen SE, et al. Genome-wide analysis of 102,084 migraine cases identifies 123 risk loci and subtype-specific risk alleles. Nat Genet 2022; 54: 152-160. 20220203. DOI: 10.1038/s41588-021-00990-0.

Reviewer 4 Report

Migraine is a debilitating pain condition, and despite the advances made for a better treatment, a significant portion of this population still are undertreated. The present article provide an overview of the N-methyl-D-aspartate receptor as a potential target for migraine, in both pre-clinical and clinical studies. The review is well written and provides useful information for pain researchers. I have only a few minor comments:

In part 3 Ion channels and migraine pathophysiology, the authors could use some more recent references with regard the channelopathies in FHM (e.g. Pietrobon D, 2013 Biochim Biophys Act). The authors could expand in regard if mutations/missense variations/polymorphisms in the NMDA receptor are described in migraine patients. 

In part 5 NMDA receptor, line 138, the authors could include the findings by Furukawa et al., 2005 Nature.

In 5.1 NMDAR in migraine, is missing a reference for the expression of the receptor in the trigeminalvascular neurons (line 175). Also, line 180, the name of the author is incorrect (Guerrero-Toro instead of Guerro-Toro)

in 5.3 NDMA antagonists in clinical studies, from line 248 to 258 is missing references. 

The authors summarize main findings in Boxes 1 and 2, but it is never mentioned in the text. Should include in the text.

Figure 2. the authors should use the conventional nomenclature for the receptor (N-Methyl D-aspartate instead of N-methylate D-aspartate).

I highly suggest the authors to include a Perspective section. Although they described well the findings in pre-clinical and clinical studies, they mentioned in 5.3 that NMDAR antagonists have limited efficacy. The authors should discuss why NMDAR antagonists could still be a good target despite their low efficacy and potent side effect (such as ketamine abuse and addiction).

Author Response

Reviewer #4
Comments for the Author

Migraine is a debilitating pain condition, and despite the advances made for a better treatment, a significant portion of this population still are undertreated. The present article provide an overview of the N-methyl-D-aspartate receptor as a potential target for migraine, in both pre-clinical and clinical studies. The review is well written and provides useful information for pain researchers. I have only a few minor comments:

  1. In part 3 Ion channels and migraine pathophysiology, the authors could use some more recent references with regard the channelopathies in FHM (e.g. Pietrobon D, 2013 Biochim Biophys Act). The authors could expand in regard if mutations/missense variations/polymorphisms in the NMDA receptor are described in migraine patients. 

Response: Thank you for your comment. Recent references regarding channelopathies and FHM are added to section 3. So far, no mutation has been found in the NMDA receptors 6.

  1. In part 5 NMDA receptor, line 138, the authors could include the findings by Furukawa H, Singh SK, Mancusso R, et al. Subunit arrangement and function in NMDA receptors.

Response: This is added.

  1. In 5.1 NMDAR in migraine, is missing a reference for the expression of the receptor in the trigeminalvascular neurons (line 175). Also, line 180, the name of the author is incorrect (Guerrero-Toro instead of Guerro-Toro).

Response: This is added, and the name is corrected.

  1. in 5.3 NMDA antagonists in clinical studies, from line 248 to 258 is missing references. 

Response: Reference is added.

  1. The authors summarize main findings in Boxes 1 and 2, but it is never mentioned in the text. Should include in the text.

Response: This is included in the text.  

  1. Figure 2. the authors should use the conventional nomenclature for the receptor (N-Methyl D-aspartate instead of N-methylate D-aspartate).

Response: Thank you for your comment. The conventional nomenclature N-Methyl D-aspartate is added to Figure 2.

  1. I highly suggest the authors to include a Perspective section. Although they described well the findings in pre-clinical and clinical studies, they mentioned in 5.3 that NMDAR antagonists have limited efficacy. The authors should discuss why NMDAR antagonists could still be a good target despite their low efficacy and potent side effect (such as ketamine abuse and addiction).

Response: Perspective section is added.

References

  1. O'Brien M and Cairns BE. Monosodium glutamate alters the response properties of rat trigeminovascular neurons through activation of peripheral NMDA receptors. Neuroscience 2016; 334: 236-244. 20160811. DOI: 10.1016/j.neuroscience.2016.08.007.
  2. Wang XM and Mokha SS. Opioids modulate N-methyl-D-aspartic acid (NMDA)-evoked responses of trigeminothalamic neurons. J Neurophysiol 1996; 76: 2093-2096. 1996/09/01. DOI: 10.1152/jn.1996.76.3.2093.
  3. Baad-Hansen L, Cairns B, Ernberg M, et al. Effect of systemic monosodium glutamate (MSG) on headache and pericranial muscle sensitivity. Cephalalgia 2010; 30: 68-76. DOI: 10.1111/j.1468-2982.2009.01881.x.
  4. Shimada A, Cairns BE, Vad N, et al. Headache and mechanical sensitization of human pericranial muscles after repeated intake of monosodium glutamate (MSG). J Headache Pain 2013; 14: 2. 20130124. DOI: 10.1186/1129-2377-14-2.
  5. Al-Karagholi MA, Hansen JM, Abou-Kassem D, et al. Phase 1 study to access safety, tolerability, pharmacokinetics, and pharmacodynamics of kynurenine in healthy volunteers. Pharmacol Res Perspect 2021; 9: e00741. DOI: 10.1002/prp2.741.
  6. Hautakangas H, Winsvold BS, Ruotsalainen SE, et al. Genome-wide analysis of 102,084 migraine cases identifies 123 risk loci and subtype-specific risk alleles. Nat Genet 2022; 54: 152-160. 20220203. DOI: 10.1038/s41588-021-00990-0.

Round 2

Reviewer 3 Report

The authors answered my concerns and improved their manuscript.